# Using association rule mining to jointly detect clinical features and differentially expressed genes related to chronic inflammatory diseases

Rosana Veroneze[1]*, Sâmia Cruz Tfaile Corbi[2], Bárbara Roque da Silva[2], Cristiane de S. Rocha[3], Cláudia V. Maurer-Morelli[3], Silvana Regina Perez Orrico[4,5], Joni A. Cirelli[4], Fernando J. Von Zuben[1], Raquel Mantuaneli Scarel-Caminaga[2]

**1** Department of Computer Engineering and Industrial Automation, School of Electrical and Computer Engineering, University of Campinas (UNICAMP), Campinas, SP, Brazil, **2** Department of Morphology, Genetics, Orthodontics and Pediatric Dentistry, School of Dentistry at Araraquara, São Paulo State University (UNESP), Araraquara, SP, Brazil, **3** Department of Medical Genetics and Genomic Medicine, University of Campinas (UNICAMP), Campinas, SP, Brazil, **4** Department of Diagnosis and Surgery, School of Dentistry at Araraquara, São Paulo State University (UNESP), Araraquara, SP, Brazil, **5** Advanced Research Center in Medicine, Union of the Colleges of the Great Lakes (UNILAGO), São José do Rio Preto, SP, Brazil

* veroneze@dca.fee.unicamp.br, rveroneze@gmail.com

**Data Availability Statement:** Genomics data are now available at the NCBI repository: https://www.

## Abstract

### Objective

It is increasingly common to find patients affected by a combination of type 2 diabetes mellitus (T2DM), dyslipidemia (DLP) and periodontitis (PD), which are chronic inflammatory diseases. More studies able to capture unknown relationships among these diseases will contribute to raise biological and clinical evidence. The aim of this study was to apply association rule mining (ARM) to discover whether there are consistent patterns of clinical features (CFs) and differentially expressed genes (DEGs) relevant to these diseases. We intend to reinforce the evidence of the T2DM-DLP-PD-interplay and demonstrate the ARM ability to provide new insights into multivariate pattern discovery.

### Methods

We utilized 29 clinical glycemic, lipid and periodontal parameters from 143 patients divided into five groups based upon diabetic, dyslipidemic and periodontal conditions (including a healthy-control group). At least 5 patients from each group were selected to assess the transcriptome by microarray. ARM was utilized to assess relevant association rules considering: (i) only CFs; and (ii) CFs+DEGs, such that the identified DEGs, specific to each group of patients, were submitted to gene expression validation by quantitative polymerase chain reaction (qPCR).

ncbi.nlm.nih.gov/geo/query/acc.cgi?acc=GSE156993.

**Funding:** RV and FJVZ are supported by São Paulo Research Foundation (FAPESP - http://www.fapesp.br/) Grant 2017/21174-8, Coordination of Superior Level Staff Improvement (CAPES - https://www.capes.gov.br/) and Brazilian National Council for Scientific and Technological Development (CNPq - http://www.cnpq.br/) Grant 307228/2018-5. RMSC is supported by FAPESP Grants 2007/08362-8, 2009/16233-9, 2010/10882-2, 2014/16148-0 and 2016/25418-6, CAPES and CNPq Grant 304570/2017-6. The funders had no role in study design, data collection and analysis, decision to publish, or preparation of the manuscript.

**Competing interests:** The authors have declared that no competing interests exist.

## Results

We obtained 78 CF-rules and 161 CF+DEG-rules. Based on their clinical significance, Periodontists and Geneticist experts selected 11 CF-rules, and 5 CF+DEG-rules. From the five DEGs prospected by the rules, four of them were validated by qPCR as significantly different from the control group; and two of them validated the previous microarray findings.

## Conclusions

ARM was a powerful data analysis technique to identify multivariate patterns involving clinical and molecular profiles of patients affected by specific pathological panels. ARM proved to be an effective mining approach to analyze gene expression with the advantage of including patient's CFs. A combination of CFs and DEGs might be employed in modeling the patient's chance to develop complex diseases, such as those studied here.

## Introduction

As a metabolic disorder, diabetes mellitus (DM) is caused either by a deficiency of insulin's mechanism of action, by an insulin secretion deficit, or by both [1]. As recently reported by Jeong et al. [2], the prevalence of DM has increased exponentially in recent decades, being expected to affect 693 million patients within 25 years. Of all adults newly diagnosed with DM, more than 90% are affected by type 2 diabetes mellitus (T2DM) [3]. According to Jeong et al. [2], in 2017 the estimated total global healthcare expenditure considering DM was USD 850 billion, with a relevant proportion of these costs arising from the treatment of various complications associated with the progression of DM. Over a period of years most T2DM patients progress to three major groups of complications: microvascular, macrovascular, and miscellaneous [4]. Regarding miscellaneous T2DM complications, Jeong et al. [2] recently reported that dyslipidemia had the highest relative incidence risk of comorbidities that evolved after a diagnosis of T2DM in Koreans. In 2010, the third cause of premature deaths (before the age of 70 years) in Brazilian subjects was regarded as diabetes, with high fasting plasma glucose and high body mass index (BMI) being some of the major risk factors related to diabetes mortality (53,353 individuals, or 12%) [5].

Dyslipidemia (DLP) is a metabolic dysfunction that results from an increased level of lipoproteins in the blood [6, 7]. Some studies have revealed that DLP could be one factor associated with DM-induced immune cell alterations [7–9]. It is believed that pro-inflammatory cytokines produce an insulin resistance syndrome similar to that observed in DM [7, 9]. Findings concerning chronically elevated levels of inflammatory markers suggest that poor glycemic control of T2DM patients could increase risk for cardiovascular disease and infectious diseases, including periodontitis [8, 10].

Periodontitis (PD) is a common chronic inflammatory disease characterized by destruction of the periodontium, which is the supporting structures of the teeth, such as gingiva, periodontal ligament and alveolar bone [11]. PD is a microbially induced oral disease, in which the bacterial biofilm is formed on the surfaces of teeth providing a chronic microbial stimulus that elicits a local inflammatory response in the gingival tissues [12]. PD is also considered an inflammatory disorder influenced by factors such as genetics [13], immune system reactions, smoking [14] and the occurrence of systemic diseases, including DM [15]. Periodontal infection and DM have a two-way relationship [16] and PD can be recognized as the sixth largest

complication associated with DM [17]. In response to bacterial products after periodontium infection, there are local and systemic elevations of pro-inflammatory cytokines [18], which may induce alterations in the metabolism of lipids, contributing to DLP in these patients [7, 9]. Some studies indicate an association between elevation in blood lipoproteins and alterations in the periodontal condition [6, 19–21].

Currently, the interplay of T2DM, DLP and PD has been increasingly affecting patients worldwide. Those are chronic inflammatory diseases, including systemic T2DM and DLP, while PD is localized at the periodontium of the patient. Growing evidence indicates a biological connection among T2DM-DLP-PD, demonstrated by the finding that these patients present a hyperinflammatory state promoted by systemically increased levels of pro-inflammatory molecules, as reviewed by Soory et al. [22]. Moreover, all of them are considered chronic and complex diseases, since they are caused by a combination of genetic, environmental and lifestyle factors [23]. Therefore, more studies focused on detecting unknown relationships in datasets of diseased patients will contribute to a better understanding of the interplay of T2DM, DLP and PD.

Association rule mining (ARM) has been widely used to discover hidden relationships established by multiple attributes that characterize a complex process under investigation. It has several applications in the medical domain (for instance, see [24–26]) promoting highly interpretable explanations without requiring data mining expertise [27]. In addition to interpretability, another reason that makes ARM a widely used data mining technique is that the obtained rules are capable of summarizing the joint impact of several factors [27, 28]. Thus, ARM is a powerful technique to assess the supposed interplay of T2DM, DLP and PD.

The ARM was previously used to assess the T2DM survival risk [29], and to determine the T2DM comorbidities in large amounts of clinical data [30]. Ramezankhani et al. [31] showed that ARM is a useful approach to determine the most frequent subsets of attributes in people who will develop diabetes. However, this is the first study using ARM to simultaneously identify the potential clinical patterns and genetic markers of this group of diseases, thus revealing clinical features and differentially expressed genes capable of properly characterizing these chronic inflammatory diseases.

The outline of this paper is as follows. Section Materials and Methods presents the literature review and our proposed methodology. Section Results and Discussion presents the experimental results and an analytical explanation of their implications, followed by concluding remarks in Section Conclusion.

## Materials and methods

### Datasets

**Studied population.**  This research was approved by the Ethics in Human Research Committee of School of Dentistry at Araraquara (UNESP; Protocol number 50/06). Patients who voluntarily sought dental treatment at the School of Dentistry at Araraquara (UNESP), Brazil, were informed about the aims and methods of the study, providing their written consent to participate; therefore, the whole study was conducted according to the ethical principles of the Declaration of Helsinki.

The patients were characterized by the following criteria: age from 35 to 60 years, presence of at least 15 natural teeth and similar socioeconomic level. Pre-selected patients, according to their medical history, had their glycemic and lipid profiles investigated by biochemical blood analysis, and were submitted to full periodontal examination. Then, 143 patients were divided into five groups based upon diabetic, dyslipidemic and periodontal conditions:

1. Group 1: poorly controlled T2DM with DLP and PD. Number of subjects = 28.

2. Group 2: well-controlled T2DM with DLP and PD. Number of subjects = 29.

3. Group 3: DLP and PD. Number of subjects = 29.

4. Group 4: systemically healthy individuals with PD. Number of subjects = 29.

5. Group 5: systemically and periodontally healthy individuals (control group). Number of subjects = 28.

No patient in those five groups presented: history of antibiotic therapy in the previous 3 months and/or nonsteroidal anti-inflammatory drug therapy in the previous 6 months, pregnancy or use of contraceptives or any other hormone, current or former smoking addiction, history of anemia, periodontal treatment or surgery in the preceding 6 months, use of hypolipidemic drugs such as statins or fibrates, and history of diseases that interfere with lipid metabolism, such as hypothyroidism and hypopituitarism.

Additionally, patients enrolled in this study were previously investigated regarding malonaldehyde (MDA) quantification and some inflammatory cytokine levels [32], micronuclei frequency (DNA damage evaluation) [33] and lipid peroxidation [32]. In these previous studies, power analysis based on a pilot study determined that at least 20 patients in each group would be sufficient to assess differences in those molecules with 90% power and 95% confidence interval.

**Biochemical, physical and periodontal evaluations.**   Clinical criteria to include each patient in the studied group are presented in what follows. Subjects were submitted to physical and anthropometric examination for evaluating obesity such as abdominal circumference (cm), height (m), weight (kg), waist (cm), hip (cm) and body mass index [33].

After a 12-hour overnight fast, each subject was referred to a clinical analysis laboratory that collected a blood sample for evaluating: glycated haemoglobin (HbA1c) by enzymatic immunoturbidimetry, fasting plasma glucose (mg/dL) by the modified Bondar & Mead method, high-sensitivity C-reactive protein by the nephelometric method and insulin levels by the chemiluminescence method (U/L). The homeostasis model assessment (HOMA) was evaluated to calculate insulin resistance (IR). The diagnosis of T2DM was made by an endocrinologist who monitored the glycemic levels of each patient by evaluation of HbA1c; being patients considered poorly controlled (HbA1c $\geq$8.0%) or well-controlled (HbA1c $\leq$7.0%). Normoglycemic (nondiabetic) individuals presented fasting glucose levels <100 mg/dL and HbA1c <5.7% [34–36].

The lipid profile [triglycerides (TG), total cholesterol (TC), and high density lipoprotein (HDL)] was performed by enzymatic methods. Low density lipoprotein (LDL) was determined by the Friedewald formula. Individuals with transitory DLP were not included here by considering the highest cutoff values: TC $\geq$240 mg/dL, LDL $\geq$160 mg/dL, HDL <40 mg/dL, and TGs $\geq$200 mg/dL, according to the 2018 AHA / ACC / AACVPR / AAPA / ABC / ACPM / ADA / AGS / APhA / ASPC / NLA / PCNA Guideline on the Management of Blood Cholesterol [37]. It was also considered in this analysis the non-HDL-cholesterol (N-HDL-C), given by N-HDL-C = TC—HDL, being the abnormal cutoff value $\geq$130 mg/dL, which is considered to be a good predictor of cardiovascular disease (CVD) risk [38].

Diagnosis of periodontitis in at least 4 non-adjacent teeth, including local signs of inflammation, loss of the connective tissue attachment of gingiva to teeth (clinical attachment loss, CAL $\geq$4mm), and tissue destruction (presence of deep periodontal pockets $\geq$6mm) was adopted according to the American Academy of Periodontology [39]. Each subject underwent a periodontal clinical examination performed at 6 sites per tooth. The presence of deep

periodontal pockets ≥6mm with CAL ≥5mm and bleeding on probing in at least 8 sites distributed in different quadrants of the dentition were the criteria of severe periodontitis [40].

Regarding the mutagenesis analysis, the description of the peripheral blood sampling, cell culture and cytokinesis-block micronucleus (CBMN) assay can be found in Corbi et al. [33].

Table 1 summarizes the clinical features collected from the 143 investigated subjects. The clinical feature dataset is available in S1 File.

**Isolation of peripheral blood mononuclear cells, RNA extraction and microarray analysis.** Patients with greater glycemic, lipid and periodontal homogeneity parameters had their transcriptome investigated (30 subjects in total) from peripheral blood mononuclear cells (PBMCs), divided into: Group 1 (number of subjects = 5), Group 2 (number of subjects = 7), Group 3 (number of subjects = 6), Group 4 (number of subjects = 6) and Group 5 (number of subjects = 6). PBMCs were isolated, and total RNA was extracted using TRizol (Invitrogen, Rockville, MD, USA) and purified by an RNeasy Protection Mini Kit (Qiagen, Hilden, Germany) according to the manufacturer's instructions. RNA was quantified by a NanoVue Spectrophotometer (GE Healthcare Life Sciences, Oslo, Norway), and its integrity was assessed by agarose gel electrophoresis (1%). Only RNA samples in the $\lambda(260/280)$ and $\lambda(260/230)$ reasons between 1.8 and 2.2 were used for microarray and quantitative real-time PCR analyses. Microarray data were generated from 500 nanograms of RNA as the initial input of each sample in the GeneChip IVT Labeling Kit and hybridized to the U133 Plus 2.0 (Affymetrix Inc., Santa Clara, CA, USA) arrays, which comprise 54,675 human transcripts. The U133 Plus 2.0 arrays were scanned twice using the GeneChip Scanner 3000 7G (Affymetrix Inc., Santa Clara, CA, USA). The Robust Multichip Average (RMA) strategy was used to preprocess raw .CEL files [41, 42]. This strategy performs background correction through a normal-exponential convolution model, quantile normalizes the probe intensities and summarizes them into probeset-level quantities using an additive model fit through the median-polish strategy [43]. The gene expression dataset is available in S2 File.

## Association rule mining

Let $\mathbf{A}_{n \times m}$ be a binary data matrix with the row index set $X = \{1, 2, \ldots, n\}$ and the column index set $Y = \{1, 2, \ldots, m\}$. Each row represents a *transaction*, and each column represents an *item*. Each element $a_{ij} \in \mathbf{A}$ holds the binary relationship between transaction $i$ and item $j$. Let $(X, Y)$ denote the entire matrix $\mathbf{A}$ and $(I, J)$ denote a submatrix of $\mathbf{A}$ with $I \subseteq X$ and $J \subseteq Y$.

**Definition 1** *A subset* $J = \{j_1, \ldots, j_s\} \subseteq Y$ *is called an itemset.*

For a subset $J \subseteq Y$, we define $J^{\downarrow} = \{x \in X | a_{xj} = 1, \forall j \in J\}$ as the set of transactions common to all the items in $J$. The support of an itemset $J$ is given by $\sigma(J) = |J^{\downarrow}|$.

The problem of mining all frequent itemsets can be described as follows: determine all subsets $J \subseteq Y$ such that $\sigma(J) \geq minSup$, where *minSup* is a user-defined parameter.

To reduce the computational cost of the frequent itemset (pattern) mining problem, some algorithms mine only the *maximal frequent itemsets*, i.e., those frequent itemsets from which all supersets are infrequent and all subsets are frequent. The problem of this approach is that it leads to loss of information since the supports of the subsets of the maximal frequent itemsets are not available. An option to reduce the computational cost of the frequent pattern mining problem without loss of information is to mine only the *closed frequent itemsets*. A frequent itemset $J$ is called *closed* if there exists no superset $H$ ǎŠf $J$ with $H^{\downarrow} = J^{\downarrow}$. Remarkably, the set of closed frequent itemsets uniquely determines the exact frequency of all frequent itemsets, and it can be orders of magnitude smaller than the set of all frequent itemsets [44]. Therefore, this approach drastically reduces the number of rules that have to be presented to the user, without any information loss [45].

**Table 1. Description of the clinical features of the 143 subjects enrolled in this study (%*ts* stands for % of tooth sites).**

| Characteristic | # | Attribute | Alias | Unit | Domain |
|---|---|---|---|---|---|
| Demographic | 1 | Sex | Sex | | 1. Female |
| | | | | | 2. Male |
| | 2 | Age | Age | yr. | 1. ≤50 |
| | | | | | 2. >50 |
| Cardiovascular and obesity risk | 3 | Body Mass Index | BMI | m/kg$^2$ | 1. Underweight: <18.5 |
| | | | | | 2. Normal weight: [18.5, 25) |
| | | | | | 3. Overweight: [25, 30) |
| | | | | | 4. Obesity class I: [30, 35) |
| | | | | | 5. Obesity class II e III: ≥35 |
| | 4 | Waist / Hip Ratio | WHR | cm/cm | (see Table 2) |
| | 5 | Abdominal Circumference | AC | cm | (see Table 3) |
| Type 2 Diabetes Mellitus | 6 | Fasting Plasma Glucose | FPG | mg/dL | 1. Normoglycemic: <100 |
| | | | | | 2. Prediabetes or high-risk: [100, 126) |
| | | | | | 3. Established diabetes: ≥126 |
| | 7 | Insulin | INS | U/L | 1. Normal: ≤25 |
| | | | | | 2. Altered: >25 |
| | 8 | Glycated Haemoglobin | HbA1c | % | 1. Normoglycemic: <5.7 |
| | | | | | 2. Prediabetes or high-risk: [5.7, 6.5) |
| | | | | | 3. Decompensation (transitory): [6.5, 8) |
| | | | | | 4. Decompensation (defined): ≥8 |
| | 9 | HOMA-IR | HOMA-IR | | 1. Normal: ≤2.15 |
| | | | | | 2. Altered: >2.15 |
| Dyslipidemia | 10 | Total Cholesterol | TC | mg/dL | 1. <150 (Optimal) |
| | | | | | 2. [150, 200) |
| | | | | | 3. [200, 240) |
| | | | | | 4. ≥240 |
| | 11 | HDL cholesterol | HDL | mg/dL | 1. <40 (Low) |
| | | | | | 2. [40, 60] |
| | | | | | 3. >60 |
| | 12 | LDL cholesterol | LDL | mg/dL | 1. <100 (Optimal) |
| | | | | | 2. [100, 130) |
| | | | | | 3. [130, 160) |
| | | | | | 4. [160, 190) |
| | | | | | 5. ≥190 |
| | 13 | Triglycerides | TG | mg/dL | 1. <150 (Optimal) |
| | | | | | 2. [150, 200) |
| | | | | | 3. ≥200 |
| | 14 | Non-HDL-Cholesterol | N-HDL-C | mg/dL | 1. <130 (Optimal) |
| | | | | | 2. [130, 160) |
| | | | | | 3. [160, 190) |
| | | | | | 4. [190, 220) |
| | | | | | 5. ≥220 |

(*Continued*)

**Table 1.** (Continued)

| Characteristic | # | Attribute | Alias | Unit | Domain |
|---|---|---|---|---|---|
| Periodontal | 15 | Visible Plaque | VP | %ts | 1. Low: <30 |
| | | | | | 2. Medium: [30, 50] |
| | | | | | 3. High: >50 |
| | 16 | Gingival Index bleeding | GI | %ts | 1. Low: <30 |
| | | | | | 2. Medium: [30, 50] |
| | | | | | 3. High: >50 |
| | 17 | Bleeding on probing | BOP | %ts | 1. Low: <30 |
| | | | | | 2. Medium: [30, 50] |
| | | | | | 3. High: >50 |
| | 18 | Total Number of Teeth | TNT | | 1. low number teeth: ≤20 |
| | | | | | 2. high number teeth: >20 |
| | 19 | Interproximal periodontal pocket depth (PPDi) ≤3mm | PPDi3mm | %ts | 1. Low: <30 |
| | | | | | 2. Medium: [30, 50] |
| | | | | | 3. High: >50 |
| | 20 | PPDi = 4—5mm | PPDi4-5mm | %ts | 1. Low: <30 |
| | | | | | 2. Medium: [30, 50] |
| | | | | | 3. High: >50 |
| | 21 | PPDi ≥ 6mm | PPDi6mm | %ts | 1. Low: <30 |
| | | | | | 2. Medium and High: ≥30 |
| | 22 | Interproximal clinical attachment loss (CALi) ≤2mm | CALi2mm | %ts | 1. Low: <30 |
| | | | | | 2. Medium: [30, 50] |
| | | | | | 3. High: >50 |
| | 23 | CALi = 3-4mm | CALi3-4mm | %ts | 1. Low: <30 |
| | | | | | 2. Medium: [30, 50] |
| | | | | | 3. High: >50 |
| | 24 | CALi ≥ 5mm | CALi5mm | %ts | 1. Low: <30 |
| | | | | | 2. Medium: [30, 50] |
| | | | | | 3. High: >50 |
| | 25 | Suppuration | SUPP | %ts | 1. Absence: <1 |
| | | | | | 2. Moderate: [1, 16) |
| | | | | | 3. Severe: ≥16 |
| Mutagenesis | 26 | Nuclear Division Index | NDI | | 1. Low: <1.87 |
| | | | | | 2. Moderate: [1.87, 2.08) |
| | | | | | 3. High: ≥2.08 |
| | 27 | Frequency of Binucleated cells with Micronuclei | MNCF | % | 1. Low: <3.05 |
| | | | | | 2. Moderate: [3.05, 7.2) |
| | | | | | 3. High: ≥7.2 |
| | 28 | Micronucleus Frequency | MNF | % | 1. Low: <3.5 |
| | | | | | 2. Moderate: [3.5, 6.1) |
| | | | | | 3. High: ≥6.1 |
| | 29 | Frequency of Nucleoplasmic Bridges | FNB | % | 1. Low: <1.21 |
| | | | | | 2. Moderate: [1.21, 2.7) |
| | | | | | 3. High: ≥2.7 |

**Definition 2** *An association rule (AR) is an expression of the form $J \Rightarrow H$, where $J$ and $H$ are itemsets, $H \cap J = \emptyset$. $J$ is called antecedent (or head) and $H$ is called consequent (or tail) of the rule.*

The support of an association rule $J \Rightarrow H$ is the number of transactions that contain the itemset $J \cup H$: $\sigma(J \Rightarrow H) = \sigma(J \cup H)$. The *confidence* of an association rule $J \Rightarrow H$ measures its predictive accuracy and is given by $conf(J \Rightarrow H) = \sigma(J \Rightarrow H)/\sigma(J)$. A rule is considered a *strong rule* if $conf(J \Rightarrow H) \geq minConf$, where *minConf* is a user-defined parameter. The *completeness* (or *recall*) is given by $comp(J \Rightarrow H) = \sigma(J \Rightarrow H)/\sigma(H)$. Remark that confidence and completeness are not symmetric measures because by definition they are conditional on the antecedent and consequent, respectively. The metric lift measures the degree of surprise of a rule and is given by $lift(J \Rightarrow H) = \sigma(J \Rightarrow H)/(\sigma(J) \times \sigma(H))$.

A user can be interested in a more specific set of association rules, where the consequents of the rules describe a target attribute. These rules are known as class association rules (CARs).

**Definition 3** *A class association rule (CAR) is an expression of the form $J \Rightarrow c$, where $J$ is an itemset and $c$ is a class label (a target item).*

In this work, each *item* is given by an attribute-value pair. Thus, for instance, FPG = 3 is an item; {AC = 3, FPG = 3, HbA1c = 4} is an itemset; and {AC = 3, FPG = 3, HbA1c = 4} ⇒ {GI = 3, BOP = 3} is an association rule.

Given that the result to be presented to the user is more parsimonious, we will focus on closed frequent itemsets here. The patterns will be mined using the RIn-Close_CVCP algorithm [46, 47], which is a fast algorithm and avoids the necessity of the *itemization step* [47]. Its implementation is available at https://github.com/rveroneze/rinclose.

**Association rule mining from the clinical features alone.** T2DM, DLP, and PD have their own specific characteristics (features or attributes) generally taken as decision variables to perform a diagnosis. However, given the increasing incidence of patients affected by different interplays of T2DM-DLP-PD, we originally used ARM to assess whether there are joint attributes present in patients with these comorbidities that might indicate the biological interrelationship among them.

Fig 1 shows a flowchart that summarizes the process of association rule mining from the dataset containing solely clinical features. From the clinical features collected from the investigated patients (presented in Table 1), we selected the most clinically relevant to diagnose T2DM, DLP and PD diseases isolated. We did not use the mutagenesis attributes because they are not applied in a clinical routine for disease diagnosis. The following 17 clinical features were selected for this analysis: BMI, WHR, AC, FPG, HbA1c, HOMA-IR, TC, HDL, LDL, TG, N-HDL-C, GI, BOP, PPDi6mm, CALi34mm, CALi5mm and SUPP. Thus, the dataset to be analyzed has 143 subjects and 17 attributes. BMI, WHR and AC attributes represent characteristics that confer cardiovascular and obesity risk, according to the World Health Organization [19, 48]. The N-HDL-C attribute is considered a good predictor of CVD risk [38]. The glycemic parameters: FPG, HbA1c and HOMA-IR (Homeostasis Model Assessment to calculate the insulin resistance) are considered essential for the diagnosis of T2DM and its metabolic control [35, 36]. TC, HDL, LDL and TG are important lipid parameters to diagnose DLP [37]. Regarding periodontitis, the American Academy of Periodontology (AAP) utilizes the clinical periodontal parameters: GI, BOP, PPDi6mm, CALi3-4mm, CALi5mm and SUPP [39, 40].

The parameters used in ARM were: *minSup* = 14 and *minConf* = 70%. A rule was considered interesting whenever at least one of the following attributes is present: PPDi6mm = 2; GI, BOP, CALi34mm, CALi5mm, SUPP ∈{2, 3}. We followed those clinical periodontal parameters, as recommended by the AAP, because they indicate periodontal disease activity. Those selected attributes are considered relevant to identify individuals undoubtedly affected by moderate or severe periodontitis, allowing us to check if there is an evident association

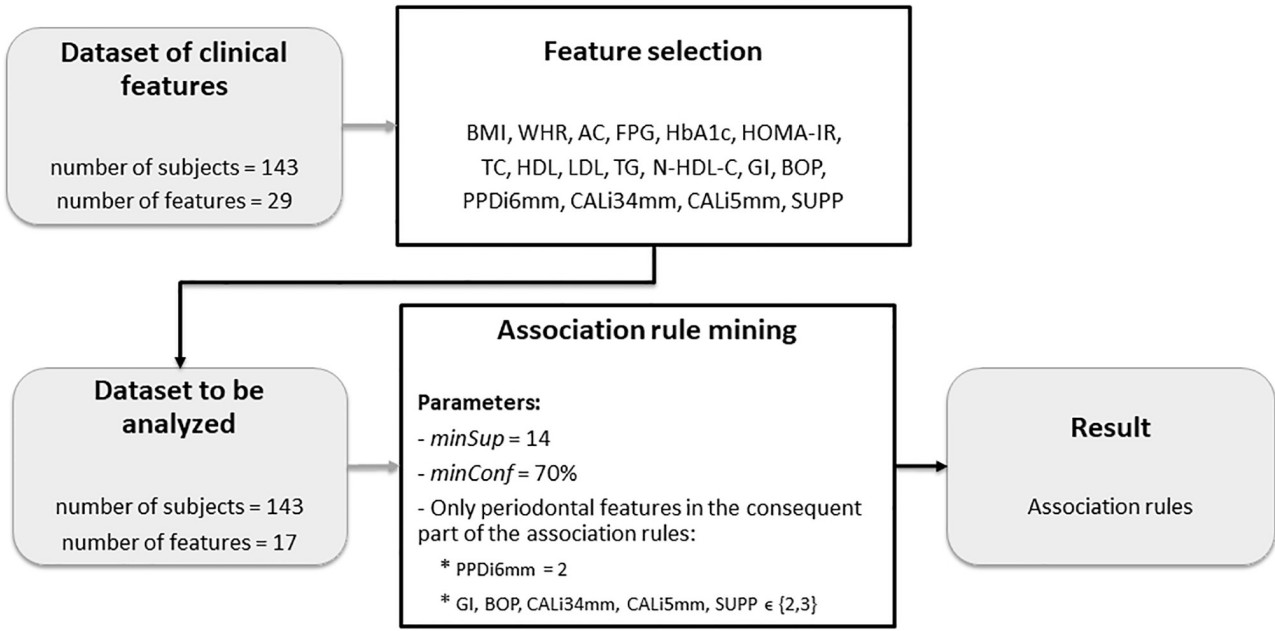

**Fig 1. Flowchart that summarizes the process of association rule mining from the dataset containing solely clinical features.**

between both systemic diseases (T2DM and DLP) and PD. In this way, we corroborate the existence of a T2DM-DLP-PD biological interrelationship.

In addition, we performed an analysis focusing on the cardiovascular and obesity risk attributes to determine whether they are associated with periodontal disease. Therefore, we performed an analysis with only the cardiovascular and obesity risk attributes in the antecedent part of the rule (BMI, WHR, AC, FPG, N-HDL-C), and the same attributes in the consequent part of the rule. We also performed an analysis comprising only T2DM patients presenting diabetic dyslipidemia, which are the 10 patients from Groups 1 and 2 having TG $\geq$204 mg/dL and HDL <38 mg/dL [49, 50].

The results of these analysis will be presented and discussed in Section Results and Discussion.

**Association rule mining from the clinical features and gene expression datasets in conjunction.** The transcriptome of the patients studied here obtained from PBMCs by microarray was analyzed utilizing bioinformatics and statistical tools, as described in topic Isolation of peripheral blood mononuclear cells, RNA extraction and microarray analysis. Those analyses, developed as regularly, produced a list of differentially expressed genes (DEGs). However, in that kind of analysis the gene expression profile obtained by the probesets did not consider the patient's clinical features (CFs). In conventional bioinformatics and statistical tools, adequate clinical diagnosis of each group of patients is used to determine whether a DEG is related to a specific pathological condition. Here, we used ARM to identify the joint interplay of CFs and DEGs, having the advantage of taking together CFs and genetic markers to identify each combination of T2DM-DLP-PD complex diseases. This approach might contribute to better identifying new targets for the diagnosis of each combination of those complex diseases, as well as for modeling the patient's chance to develop them.

Fig 2 shows a flowchart that summarizes the process of class association rule mining from the dataset containing both CFs and DEGs. First, we performed the preprocessing of the

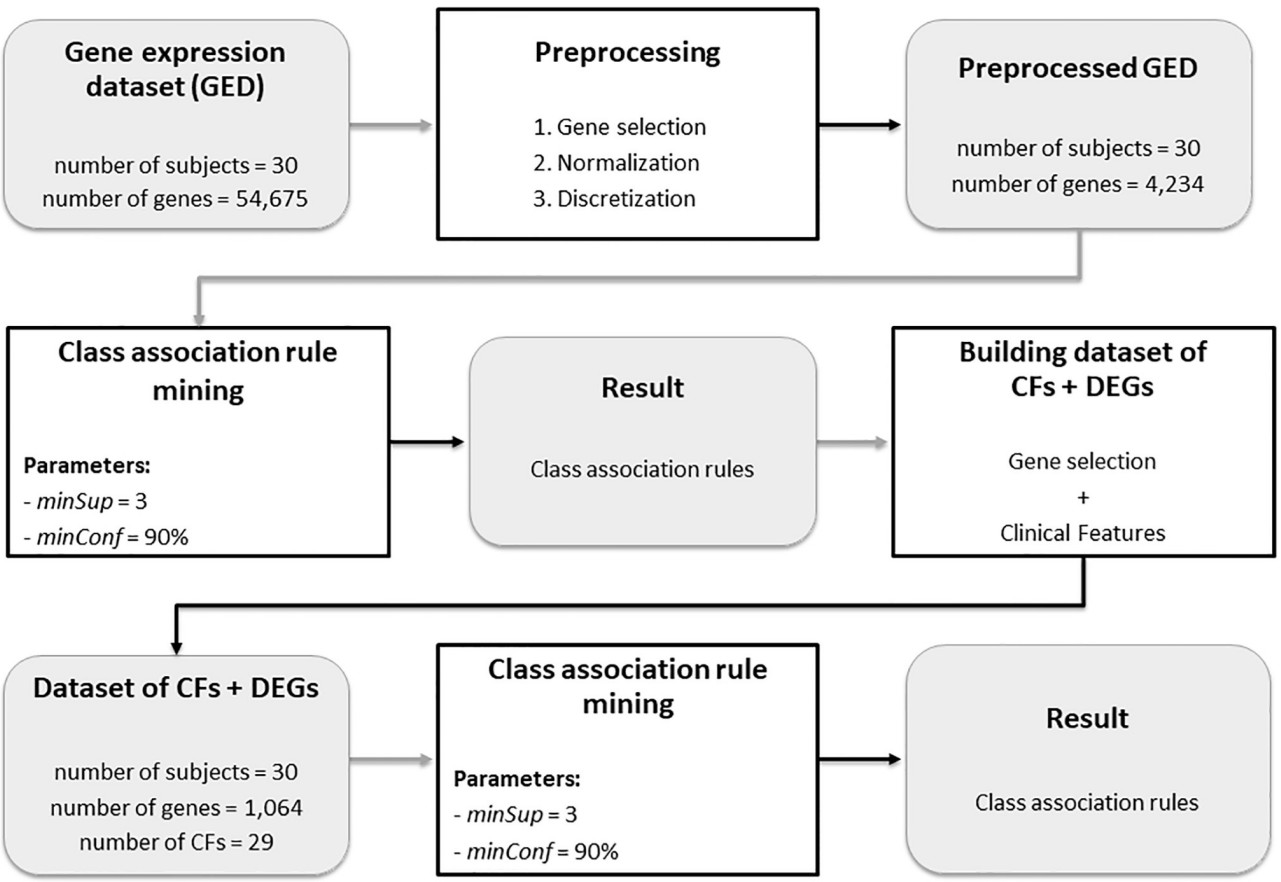

**Fig 2. Flowchart that summarizes the process of class association rule mining from the dataset containing both clinical features (CFs) and differentially expressed genes (DEGs).**

original gene expression dataset (GED), which has the gene expression profile of 54,675 genes obtained from the transcriptome of the 30 subjects, in the following three steps:

1. Gene selection: we filtered out genes with small profile variance, in specific we filtered out gene expression profiles with variation less than 0.1 when considering the difference between its maximum and minimum values. It was done because gene profiling experiments typically include genes that exhibit little variation in their profile and these genes are usually uninteresting. Thus, these genes are commonly removed from the analysis. With this filter, 50.441 genes were removed, leaving 4.234 genes for the subsequent analysis.

2. Normalization: we used zero-mean normalization to adjust the values measured on different scales to a common scale. Let $\mathbf{g}$ be the gene expression profile of a gene $g$ for the 30 subjects of our study. The normalized gene expression profile $\hat{\mathbf{g}}$ is given by $\hat{\mathbf{g}} = (\mathbf{g} - \mathrm{avg}(\mathbf{g}))/\mathrm{std}(\mathbf{g})$, where $\mathrm{avg}(\mathbf{g})$ and $\mathrm{std}(\mathbf{g})$ are, respectively, the sample average and the sample standard deviation of $\mathbf{g}$.

3. Discretization: if a normalized gene expression value was above 1.0, it was considered over-expressed (and it is represented by the value 1 in our results); if a normalized gene expression value was below -1.0, it was considered under-expressed (and it is represented by the value -1 in our results); otherwise the gene expression value was considered uninteresting and was ignored.

We performed the mining of CARs in this preprocessed GED with the following parameters: *minSup* = 3 and *minConf* = 90%. The group of each individual (Groups 1 to 5) is the target attribute. The result, containing 118 CARs, was used for a new phase of gene selection as described in what follows. The 118 CARs have a coverage of 1081 genes (this means that 1081 genes are presented in these rules). Of these 1081 genes, 17 genes are present in conflicting rules, exhibiting the same value for the control group (Group 5) and for the other groups (Groups 1 to 4). Therefore, these 17 PD genes were discarded. Thus, 1081 − 17 = 1064 genes were selected for the new phase of analysis, together with the 29 CFs listed in Table 1. In this new phase of analysis, we performed the mining of CARs with the same parameters, i.e., *minSup* = 3 and *minConf* = 90%. The results will be presented and discussed in Section Results and Discussion.

## Reverse transcription-quantitative polymerase chain reaction (RT-qPCR) Real-Time Analysis

To biologically validate the genes selected from the CARs considering the CFs+DEGs, we conducted RT-qPCR analyses in all 143 patients (including the 30 patients who were analyzed by microarray) distributed into the 5 groups, according to the subitem Studied population. Reverse transcription reactions were performed utilizing the High Capacity Kit (Thermo Fisher Scientific). Complementary DNA (cDNA) was used to perform qPCR reactions for the selected DEGs, which are represented as probe sets in Table 7. To investigate the expression of the probe (or gene) identified by the rule selected for each group of patients, the TaqMan® gene expression assay specific for each of these "target" genes was utilized. Each target gene is normalized by a gene considered an endogenous control of the qPCR reactions, in this case, we utilized the *GAPDH -Glyceraldehyde-3-Phosphate Dehydrogenase* gene (Hs02758991_g1), due to its housekeeping expression pattern.

All reactions were performed in duplicate utilizing the 7500 Real-Time PCR-System (Thermo Fisher Scientific, Foster City, CA, USA). To calculate gene expression, Expression Suite Software was used (Thermo Fisher Scientific, Foster City, CA, USA), which employs the comparative Cycle Threshold ($\Delta Ct$) method for multivariate data analysis. Statistical analysis to find differences in the gene expression by the values of $2^{-\Delta Ct}$ between the groups was performed by the Mann-Whitney test, utilizing GraphPad Prism software, version 5.0, and considering a significance level of 0.05 [51].

## Results and discussion

### Association rules for the dataset of clinical features (CFs)

It was obtained 78 rules comprising the CF dataset, which are presented in S1 Table. The periodontists and geneticist experts analyzed those rules to select examples of rules of high clinical relevance to demonstrate the T2DM-DLP-PD interrelationship. To select the rules, the following requirements were established in decreasing order of relevance:

1. In the antecedent part of the rule, the joint presence of attributes with altered values in these characteristics of Tables 1, 2 and 3: cardiovascular and obesity risk; T2DM; and DLP;

2. The highest confidence value.

The rules of Table 4 present, in general, WHR = 4 and AC = 3, which represent very high cardiovascular and obesity risk for all ages of both male and female (see Tables 2 and 3); FPG = 3, HbA1c = 4 and HOMA-IR = 2 represent the worst glycemic parameters, evidencing that those patients have established T2DM with defined metabolic decompensation and

**Table 2. Waist / hip ratio domain.**

| Female | | | |
|---|---|---|---|
| | Age ≤ 39 | 39 < Age ≤ 49 | Age > 49 |
| 1. Low | <0.72 | <0.73 | <0.74 |
| 2. Moderate | [0.72, 0.79) | [0.73, 0.80) | [0.74, 0.82) |
| 3. High | [0.79, 0.84] | [0.80, 0.87] | [0.82, 0.88] |
| 4. Very High | > 0.84 | > 0.87 | > 0.88 |
| **Male** | | | |
| | Age ≤ 39 | 39 < Age ≤ 49 | Age > 49 |
| 1. Low | <0.84 | <0.88 | <0.90 |
| 2. Moderate | [0.84, 0.92) | [0.88, 0.96) | [0.90, 0.97) |
| 3. High | [0.92, 0.96] | [0.96, 1] | [0.97, 1.02] |
| 4. Very High | > 0.96 | > 1 | > 1.02 |

**Table 3. Table caption Nulla mi mi, venenatis sed ipsum varius, volutpat euismod diam.**

| | Female | Male |
|---|---|---|
| 1. Low risk | < 80 | < 94 |
| 2. High risk | [80, 88) | [94, 102) |
| 3. Very high risk | ≥ 88 | ≥ 102 |

insulin resistance; the patients are also dyslipidemic as demonstrated by the highest levels of total cholesterol (TC = 4) and triglycerides (TG = 3). The consequent part of those rules is BOP = 3, which means that more than 50% of tooth sites bleed during the periodontal exam, demonstrating wide and active inflammation of the periodontal tissues including the gingiva. There are 4 rules showing as consequent SUPP = 2, meaning that those patients have a moderate suppuration, since it affects 1% to 16% of tooth sites, indicating the presence of an established periodontitis. The seventh and eighth rules of Table 4 show TC = 4 and N-HDL-C = 5, meaning that individuals with the highest levels of TC and N-HDL-C have 78% of confidence of presenting BOP = 3 or SUPP = 2, demonstrating wide and active inflammation of the periodontal tissues and an established periodontitis.

**Table 4. Association rules for the clinical feature dataset.**

| Rule | $\sigma_{rule}$ | $\sigma_{head}$ | $\sigma_{tail}$ | %Conf. | Lift |
|---|---|---|---|---|---|
| WHR = 4, FPG = 3, HbA1c = 4, TG = 3 ⇒ BOP = 3 | 14 | 14 | 74 | 100.00 | 1.93 |
| AC = 3, FPG = 3, HbA1c = 4, HOMA-IR = 2, TG = 3 ⇒ BOP = 3 | 15 | 15 | 74 | 100.00 | 1.93 |
| AC = 3, FPG = 3, HOMA-IR = 2, TG = 3 ⇒ BOP = 3 | 22 | 26 | 74 | 84.62 | 1.64 |
| WHR = 4, AC = 3, FPG = 3, HOMA-IR = 2, TG = 3 ⇒ BOP = 3 | 19 | 23 | 74 | 82.61 | 1.60 |
| WHR = 4, FPG = 3, TG = 3 ⇒ BOP = 3 | 21 | 26 | 74 | 80.77 | 1.56 |
| WHR = 4, FPG = 3, HOMA-IR = 2, TG = 3 ⇒ BOP = 3 | 20 | 25 | 74 | 80.00 | 1.55 |
| AC = 3, HOMA-IR = 2, TC = 4, TG = 3 ⇒ SUPP = 2 | 15 | 19 | 67 | 78.95 | 1.68 |
| TC = 4, N-HDL-C = 5 ⇒ BOP = 3 | 18 | 23 | 74 | 78.26 | 1.51 |
| TC = 4, N-HDL-C = 5 ⇒ SUPP = 2 | 18 | 23 | 67 | 78.26 | 1.67 |
| AC = 3, HOMA-IR = 2, TC = 4 ⇒ SUPP = 2 | 23 | 31 | 67 | 74.19 | 1.58 |
| WHR = 4, AC = 3, HOMA-IR = 2, TC = 4 ⇒ SUPP = 2 | 18 | 25 | 67 | 72.00 | 1.54 |

**Table 5. Association rules for the clinical feature dataset—Cardiovascular risk.**

| Rule | $\sigma_{rule}$ | $\sigma_{head}$ | $\sigma_{tail}$ | %Conf. | Lift |
|---|---|---|---|---|---|
| WHR = 4, AC = 3, FPG = 3 ⇒ BOP = 3 | 24 | 28 | 74 | 85.71 | 1.66 |
| FPG = 3 ⇒ BOP = 3 | 35 | 41 | 74 | 85.37 | 1.65 |
| AC = 3, FPG = 3 ⇒ BOP = 3 | 28 | 33 | 74 | 84.85 | 1.64 |
| WHR = 4, FPG = 3 ⇒ BOP = 3 | 26 | 31 | 74 | 83.87 | 1.62 |
| N-HDL-C = 5 ⇒ BOP = 3 | 18 | 23 | 74 | 78.26 | 1.51 |
| N-HDL-C = 5 ⇒ SUPP = 2 | 18 | 23 | 67 | 78.26 | 1.67 |
| BMI = 3, WHR = 4, AC = 3 ⇒ SUPP = 2 | 18 | 24 | 67 | 75.00 | 1.60 |
| BMI = 3, WHR = 4 ⇒ SUPP = 2 | 20 | 28 | 67 | 71.43 | 1.52 |

There was interest in verifying the association of cardiovascular and obesity parameters with the presence of periodontitis. In that analysis we also included the N-HDL-C attribute, which predicts CVD risk even better than LDL [52]. The rules obtained by focusing on only those 11 attributes are presented in Table 5. We highlighted the rules: BMI = 3, WHR = 4, AC = 3 ⇒ SUPP = 2 and N-HDL-C = 5 ⇒ BOP = 3, as supporting the evidence of an association between cardiovascular risk factors and periodontitis. The obtained rules support the clear association between N-HDL-C and parameters of periodontitis. The N-HDL-C was the best predictor among all cholesterol measures, both for coronary artery disease events and for strokes [53]. More recently, this was confirmed, since the highest N-HDL-C concentrations in blood (≥220 mg/dL, which is equivalent to ≥5.7 mmol/L) were associated with the highest long-term risk of atherosclerotic cardiovascular disease [54]. Here we observed exactly this highest level of N-HDL-C in the rules of Table 5. Interestingly, there are good reasons for the usefulness of N-HDL-C in monitoring patients, since unlike LDL, N-HDL-C does not require the triglyceride concentration to be 4.5 mmol/L (400 mg/dL), and has an additional advantage of not requiring patients to fast before blood sampling. Therefore, it is certainly a better measure than calculated LDL for patients with increased plasma triglyceride concentrations [38, 53].

In general, these rules demonstrate the interplay between cardiovascular and obesity risk, T2DM, DLP and PD, which is in line with some studies as reviewed by Soory [22] and Khumaedi et al. [8]. These diseases manifest persistent elevation of systemic inflammatory mediators, characterizing chronic inflammation [8]. It is known to be one of the atherosclerosis nontraditional risk factors and has a role in every phase of atherogenesis [8]. Atherogenic dyslipidemia is expressive among T2DM individuals, for example, in 10 − 15% of the European population [49, 50]. Therefore, we performed an analysis comprising only our 10 T2DM patients presenting diabetic dyslipidemia [49, 50]. The rules found for this pathologic condition are presented in Table 6. We highlighted the rule: FPG = 3, HOMA-IR = 2, TC = 2, HDL = 1, TG = 3 ⇒ BOP = 3, as it demonstrated that diabetic dyslipidemia was associated with more than 50% of tooth sites bleeding, one of the main significant signals of periodontium

**Table 6. Association rules for the clinical feature dataset—Diabetic dyslipidemia.**

| Rule | $\sigma_{rule}$ | $\sigma_{head}$ | $\sigma_{tail}$ | %Conf. | Lift |
|---|---|---|---|---|---|
| AC = 3, FPG = 3, HOMA-IR = 2, HDL = 1, TG = 3 ⇒ GI = 3 | 6 | 6 | 6 | 100.00 | 1.67 |
| AC = 3, FPG = 3, HOMA-IR = 2, TC = 2, HDL = 1, TG = 3 ⇒ GI = 3, BOP = 3 | 5 | 5 | 5 | 100.00 | 2.00 |
| FPG = 3, HOMA-IR = 2, TC = 2, HDL = 1, TG = 3 ⇒ BOP = 3 | 6 | 6 | 6 | 100.00 | 1.67 |
| AC = 3, FPG = 3, HOMA-IR = 2, HDL = 1, TG = 3 ⇒ GI = 3, PPDi6mm = 1 | 5 | 6 | 5 | 83.33 | 1.67 |

inflammation. Periodontitis is the most common cause of chronic inflammation in diabetic patients. Both periodontitis and diabetes have detrimental effects on each other in terms of alveolar bone destruction and poor metabolic control, by continuous inflammatory mediator activation [8].

### Association rules for the datasets of clinical features and differentially expressed genes in conjunction

Remark that we used ARM to obtain rules with joint patterns of CFs and DEGs, having the advantage of taking together the clinical characteristics and the genetic markers to identify each T2DM-DLP-PD combination of complex diseases. Also different from the rules considering only CFs (Table 4), the CF+DEG-rules were obtained for identifying specifically a group of patients. Therefore, both CFs and DEGs were considered in the antecedent part of the rules, and the consequent part of the rules is given by the number representing the groups (Groups 1 to 5). It was obtained 161 CF+DEG-rules, which are presented in S2 Table.

Because of the importance of biologically validating the CF+DEG-rules, Periodontists and Geneticist experts selected only one discriminant rule for each of the five groups, as presented in Table 7. The Periodontists and Geneticist experts make the decision of the CF+DEG-rules's choice following these criteria in decreasing order of relevance:

1. The joint presence of attributes showing values as altered as possible (according to the reference values presented in Tables 1, 2 and 3) referring to the cardiovascular and obesity risk, T2DM, DLP, PD, and also, at lower relevance, mutagenesis and demographic characteristics;

2. The presence of one probe representing an over-expressed gene, such as '229026_at = 1';

3. The highest confidence value;

4. The highest completeness value.

All the selected rules in Table 7 have 100% of confidence, which means that all subjects who give support to a rule are from the same group.

Specifically to Group 1 of patients (poorly controlled T2DM with DLP and PD), the selected rule means that 80% of the patients of Group 1 have high abdominal circumference (AC = 3), meaning high CHD risk; altered glycemic parameters (FPG = 3, HbA1c = 4, HOMA-IR = 2), evidencing that those patients have established T2DM with defined metabolic decompensation

**Table 7. Association rules for the clinical feature and gene expression datasets in conjunction.**

| Rule | %Comp. | %Conf. | Lift |
|------|--------|--------|------|
| AC = 3, FPG = 3, INS = 1, HbA1c = 4, HOMA-IR = 2, HDL = 2, TG = 3, VP = 3, BOP = 3, PPDi6mm = 1, CALi2mm = 1, SUPP = 2, 223130_s_at = -1, 229026_at = 1 ⇒ 1 | 80.00 | 100.00 | 6.00 |
| HOMA-IR = 2, TC = 4, TG = 3, N-HDL-C = 5, 208485_x_at = 1, 212386_at = -1 ⇒ 2 | 71.00 | 100.00 | 4.29 |
| FPG = 1, HDL = 2, PPDi3mm = 3, PPDi6mm = 1, MNCF = 2, 223422_s_at = 1, 224902_at = 1 ⇒ 3 | 67.00 | 100.00 | 5.00 |
| BMI = 2, FPG = 1, INS = 1, HbA1c = 1, HOMA-IR = 1, HDL = 2, TG = 1, TNT = 2, PPDi6mm = 1, CALi2mm = 1, CALi3-4mm = 3, N-HDL-C = 1, 1560999_a_at = 1, 228766_at = -1, 244413_at = 1 ⇒ 4 | 67.00 | 100.00 | 5.00 |
| Age = 1, FPG = 1, INS = 1, HbA1c = 1, TG = 1, VP = 1, GI = 1, BOP = 1, TNT = 2, PPDi3mm = 3, PPDi4-5mm = 1, PPDi6mm = 1, CALi5mm = 1, SUPP = 1, NDI = 2, MNCF = 1, MNF = 1, FNB = 1, 236395_at = 1 ⇒ 5 | 67.00 | 100.00 | 5.00 |

and insulin resistance; high triglyceride level (TG = 3); established severe periodontitis as denoted by VP = 3 (more than 50% of tooth sites showing poor oral hygiene), BOP = 3 (more than 50% of tooth sites bleeding), PPDi6mm = 1 (up to 30% of tooth sites with deep periodontal pockets), and SUPP = 2 (suppuration at maximum of 16% of tooth sites). Though the following attributes did not contribute to the identification of Group 1, they also did not disturb it: INS = 1, HDL = 2 and CALi2mm = 1.

The rule selected for Group 2 (well-controlled T2DM with DLP and PD) means that 71% of the patients of Group 2 have insulin resistance demonstrated by HOMA-IR = 2; and the highest levels of total cholesterol (TC = 4), triglycerides (TG = 3) and non-HDL-cholesterol (N-HDL-C = 5). Surprisingly, considering the first criterion for selecting these 5 rules, for identifying Group 2 of patients, a few rules were obtained. Because of this, in the selected rule there were no attributes regarding the cardiovascular and obesity risk and PD. Moreover, it should be taken into account that the rules obtained for Group 2 of patients should reflect the clinical criteria defined to select the patients. For example, in comparison with Group 1, Group 2 of patients differs only by the better metabolic control of T2DM.

The rule selected for Group 3 (DLP and PD) means that 67% of the patients have normal fasting plasma glucose (FPG = 1) which is expected since they are not affected by T2DM; they present altered HDL levels (HDL = 2), and they are affected by PD, since up to 30% of tooth sites present very deep periodontal pockets (PPDi6mm = 1). Moreover, in this rule the moderate frequency of binucleated cells with micronuclei (MNCF = 2) means that the circulating blood of the patients is affected by a moderate level of mutagenesis, probably as a consequence of the altered lipid metabolism of the patients. Indeed, a previous study of our research group enrolling the same patients showed significantly higher mRNA levels of leptin in dyslipidemic individuals (Groups 1, 2 and 3). Moreover, those leptin mRNA levels were significantly correlated with periodontal parameters such as BOP, suppuration and mainly CALi $\geq$ 5 mm [55].

Regarding Group 4 (systemically healthy individuals with PD), the selected rule means that 67% of the patients of this group are not obese, diabetic or dyslipidemic, as expected by the underlined clinical criteria for selecting them. Those patients are only affected by generalized periodontitis with pronounced alveolar bone loss, since they present more than 50% of tooth sites with 3 to 4 mm of clinical attachment loss (CALi34mm = 3), and up to 30% of tooth sites with very deep periodontal pockets (PPDi6mm = 1).

The rule selected for Group 5 (systemically and periodontally healthy individuals, or control group) means that 67% of the patients of this group are not characterized by obesity, T2DM or DLP, as expected by the underlined clinical criteria for selecting them. In addition, they did not present active PD because it was not present in the rule any domain of bleeding or inflammation, and the presence of the shallow periodontal pockets (PPDi3mm = 3) in at least 50% of tooth sites is not an indicator of periodontal disease. Conversely, the occurrence of up to 30% of tooth sites with PPDi45mm, PPDi6mm = 1, and clinical attachment loss (CALi5mm = 1) suggests that those patients were previously affected by localized PD. Moreover, although the rule includes the mutagenic parameters, their values are not altered.

To proceed to the biological validation of DEGs, we chose to validate by RT-qPCR (see Subsection Reverse transcription-quantitative polymerase chain reaction (RT-qPCR) Real-Time Analysis) one highly expressed gene in each of the five rules. Certainly, more rules with more probes/DEGs could be selected for validation, but we had limitations in the volume of the biological sample of the patients (RNA obtained from PBMCs).

For Group 1, we selected the probe 229026_at = 1, whose gene is *CDC42SE2 (Cell Division Cycle 42 Small Effector 2)*, detected by the TaqMan assay Hs00184113_m1. Although there is another gene in the rule of Group 1 (23130_s_at), this gene was down-regulated, and therefore did not meet the criteria of choice. The *CDC42SE2* gene has diverse biological functions, such

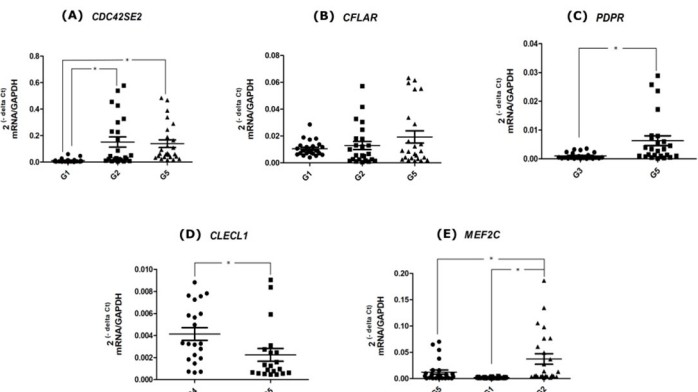

**Fig 3. Validation results by RT-qPCR of the genes considering the different Group (G) comparisons.** All mRNA levels of the investigated genes were normalized to the *GAPDH* endogenous control gene. (A) *CDC42SE2* gene expression, $^*p \leq 0.0001$; (B) *CFLAR* gene expression, no statistical difference among the groups; (C) *PDPR* gene expression, $^*p \leq 0.0002$; (D) Validation of the *CLECL1* gene expression, $^*p \leq 0.0064$; (E) Validation of the *MEF2C* gene expression, $^*p \leq 0.0425$. Data represent the mean ± SEM $2^{-\Delta Ct}$ of all patients in that group (Mann–Whitney U test; $\alpha = 5\%$).

as the organization of the actin cytoskeleton by acting downstream of *CDC42SE2*, inducing actin filament assembly, and it may play a role in early contractile events in phagocytosis in macrophages. Accordingly, the *CDC42SE2* gene alters CDC42-induced cell shape changes. In activated T-cells, the *CDC42SE2* gene may play a role in CDC42-mediated F-actin accumulation at the immunological synapse [56]. The *CDC42 (Cell Division Cycle 42)* gene encodes a small GTPase protein belonging to the Rho-subfamily, which regulates signaling pathways that control diverse cellular functions including cell morphology, migration, endocytosis and cell cycle progression [56].

In Fig 3(A), it can be observed that the *CDC42SE2* gene was down-regulated in the decompensated T2DM, dyslipidemic and PD patients (Group 1) (p-value $\leq$ 0.0001) in comparison to the healthy patients (Group 5). Actually, this finding obtained by qPCR is contrary to the expected by the rule based on the microarray data (denoted by the positive 1 value of the '229026_at'). Therefore, the qPCR method showed discordant gene expression levels from those detected by the microarray. Actually, it is not uncommon to find discrepant results of gene expression between qPCR and microarray, either because the gene expression between the diseased and control groups did not reach statistical difference or because conflicting results were found between the qPCR and microarray methods [51]. The discordant *CDC42SE2* gene expression between qPCR and microarray (not validation) means more a limitation of the method for identification of gene expression levels than a limitation of CAR mining. In addition, considering that Group 2 of patients only differs from Group 1 in patients' metabolic control, we also investigated the *CDC42SE2* gene expression in the well-controlled T2DM-DLP-PD (Group 2) patients, and we observed significantly lower levels in Group 1 but no significant difference in Group 2 in comparison to the control Group 5. Therefore, when we performed the *CDC42SE2* gene expression comparison involving Groups 1, 2 and 5, we observed the lowest expression in the worst metabolic condition of patients (Group 1), while the patients with adequate metabolic control (Group 2) had similar *CDC42SE2* expression when compared with the healthy patients of Group 5.

For Group 2, the selected probe is 208485_x_at = 1, which is the *CFLAR (CASP8 and FADD Like Apoptosis Regulator)* gene, detected by the TaqMan assay Hs01117851_m1. The protein encoded by the *CFLAR* gene is a regulator of apoptosis which may function as a crucial link

between cell survival and cell death pathways. Additionally, this protein acts as an inhibitor of TNF receptor superfamily member 6 (TNFRSF6) mediated apoptosis [56]. Considering the rule, an over-expression of the *CFLAR* gene was expected in Group 2 compared to Group 5. However, there was a similarly high expression of the *CFLAR* gene in both Groups 2 and 5 (see Fig 3(B)). We also performed the analysis of the *CFLAR* gene expression for Groups 1, 2 and 5, observing no significant difference among them, although a lower gene expression can be found in the patients with the worst metabolic condition (Group 1).

For Group 3, the rule has 2 highly expressed genes/probes, and we selected the 224902_at probe for further analysis, which is the *PDPR (Pyruvate Dehydrogenase Phosphatase Regulatory Subunit)* gene, detected by the TaqMan assay Hs01663324_m1, because it takes part in a more interesting metabolic pathway. This gene acts on the pyruvate dehydrogenase complex by catalyzing the oxidative decarboxylation of pyruvate and linking glycolysis to the tricarboxylic acid cycle and to the synthesis of fatty acids [56]. The observed significant down-regulation of the *PDPR* gene in Group 3 (DLP-PD) in comparison with the healthy Group 5 (p-value $\leq 0.0002$) by qPCR was discordant from those detected by the microarray, as shown in Fig 3(C).

Regarding Group 4 (patients affected by only PD), the rule also has 2 highly expressed genes/probes: the *IL12RB2* gene (1560999_a_at), and the *CLECL1* gene (244413_at), which was chosen to validate the gene expression by using the TaqMan assay Hs00416849_m1. The *CLECL1 (C-Type Lectin Like 1)* gene acts as a co-stimulating molecule of T cells and plays a role in the interaction of dendritic cells with T cells and the cells of the adaptive immune response [56]. In the comparison between Group 4 and Group 5, there was a highly statistically significant (p-value $\leq 0.0064$) expression of the *CLECL1* gene in Group 4, validating the DEG detected by microarray, as shown in Fig 3(D).

For Group 5 (healthy patients), the only highly expressed gene is the *MEF2C (Myocyte Enhancer Factor 2C)* gene (identified by the 236395_at probe), and detected by the TaqMan assay Hs00231149_m1. The *MEF2C* gene is involved in several normal pathways of muscular, vascular, neural, megakaryocyte and platelet development, bone marrow B lymphopoiesis, B cell survival and proliferation in response to BCR stimulation, efficient responses of IgG1 antibodies to T cell dependent antigens and normal induction of B cells from the germinal center [56]. The *MEF2C* gene expression by qPCR validated the DEG detected by microarray, as significantly highly expressed in Group 5 when compared with Group 1 (p-value $\leq 0.0425$) (see Fig 3(E)). It is interesting to compare PBMC gene expression between patients with the most opposite healthy conditions, such as Groups 1, 2 and 5, in which the worst metabolic condition (Group 1) showed the lowest level of *MEF2C* gene expression.

To our knowledge, this is the first initiative to investigate the expression of *CDC42SE2* and *CLECL1* genes in the context of T2DM, DLP and PD, demonstrating the innovative character of this study. Regarding *CFLAR* gene expression, only one study was reported in the literature investigating the relationship between body composition and BMI in children and DNA methylation. *CFLAR* gene expression was positively regulated in PBMCs of obese children [57]. Similarly, only one study investigated the *PDPR* gene with the genetic risk for DM, but the authors focused on type 1 DM, not allowing direct comparison with the T2DM results [58]. Two previous studies reported changes in the function of the MEF2C gene: Yuasa et al. [59] found MEF2C transcriptional repression in patients with T2DM, and Davegårdh et al. [60] verified a down-regulation of MEF2C related to obesity. Such results are in agreement with the findings of our study, with MEF2C being more highly expressed in patients in Group 5 (systemically and periodontally healthy individuals) than in Groups 1 and 2 (individuals with metabolic and periodontal involvement).

Although we originally utilized the ARM to investigate CFs and DEGs relevant in the context of T2DM, DLP and PD, it is important to attest that:

1. We just considered the periodontitis parameters as the consequent part of the rules because the literature demands more evidences regarding the association between systemic diseases like T2DM and DLP, with PD;

2. Regarding the CF+DEG rules, more rules could be selected for each patient group, permitting biological validation of up- or down-regulated probesets/genes, but we had limitations in the volume of biological samples of the patients (RNA obtained from PBMCs) necessary for the RT-qPCR technique.

## Conclusion

We demonstrated that ARM is a powerful data analysis technique to identify consistent patterns between the clinical and molecular profiles of patients affected by specific pathological panels. In addition, ARM was able to evidence relevant associations among important parameters of the periodontal, glycemic, lipid, cardiovascular and obesity risk conditions of the patients. Considering the qPCR validation results of the DEGs prospected by the CARs of each group of patients, four of the five genes revealed significant differences in comparison to the control group; two of them *CLECL1* and *MEF2C* genes validated the previous microarray findings. These last genes were referred to groups without systemic metabolic impairment (Group 4 and Group 5). Further studies will investigate other DEGs and other rules. Additionally, as an alternative to other commonly used techniques, ARM can be applied as a highly-interpretable mining approach to analyze the gene expression signal, with the advantage of including the patient's clinical features. Moreover, the combination of CFs and DEGs can be utilized to further estimate the patient's chance of developing complex diseases, such as those studied here.

## Supporting information

**S1 File. Clinical feature dataset.**
(CSV)

**S2 File. Gene expression dataset.**
(TXT)

**S1 Table. Association rules mined from the clinical feature dataset.**
(XLS)

**S2 Table. Class association rules mined from clinical feature and gene expression datasets in conjunction.**
(XLS)

## Author Contributions

**Conceptualization:** Rosana Veroneze, Fernando J. Von Zuben, Raquel Mantuaneli Scarel-Caminaga.

**Data curation:** Rosana Veroneze, Sâmia Cruz Tfaile Corbi, Cristiane de S. Rocha, Cláudia V. Maurer-Morelli, Silvana Regina Perez Orrico, Joni A. Cirelli, Raquel Mantuaneli Scarel-Caminaga.

**Formal analysis:** Rosana Veroneze, Sâmia Cruz Tfaile Corbi, Bárbara Roque da Silva, Cristiane de S. Rocha, Cláudia V. Maurer-Morelli, Silvana Regina Perez Orrico, Joni A. Cirelli.

**Funding acquisition:** Rosana Veroneze, Silvana Regina Perez Orrico, Fernando J. Von Zuben, Raquel Mantuaneli Scarel-Caminaga.

**Investigation:** Rosana Veroneze, Sâmia Cruz Tfaile Corbi, Bárbara Roque da Silva, Cristiane de S. Rocha.

**Methodology:** Rosana Veroneze, Cláudia V. Maurer-Morelli, Silvana Regina Perez Orrico, Fernando J. Von Zuben, Raquel Mantuaneli Scarel-Caminaga.

**Project administration:** Rosana Veroneze, Silvana Regina Perez Orrico, Fernando J. Von Zuben, Raquel Mantuaneli Scarel-Caminaga.

**Resources:** Rosana Veroneze, Cláudia V. Maurer-Morelli, Silvana Regina Perez Orrico, Fernando J. Von Zuben, Raquel Mantuaneli Scarel-Caminaga.

**Software:** Rosana Veroneze, Fernando J. Von Zuben.

**Supervision:** Cláudia V. Maurer-Morelli, Silvana Regina Perez Orrico, Joni A. Cirelli, Fernando J. Von Zuben, Raquel Mantuaneli Scarel-Caminaga.

**Validation:** Sâmia Cruz Tfaile Corbi, Cláudia V. Maurer-Morelli, Silvana Regina Perez Orrico, Joni A. Cirelli, Raquel Mantuaneli Scarel-Caminaga.

**Visualization:** Rosana Veroneze, Bárbara Roque da Silva, Silvana Regina Perez Orrico, Joni A. Cirelli, Raquel Mantuaneli Scarel-Caminaga.

**Writing – original draft:** Rosana Veroneze, Bárbara Roque da Silva, Fernando J. Von Zuben, Raquel Mantuaneli Scarel-Caminaga.

**Writing – review & editing:** Rosana Veroneze, Sâmia Cruz Tfaile Corbi, Bárbara Roque da Silva, Cristiane de S. Rocha, Cláudia V. Maurer-Morelli, Silvana Regina Perez Orrico, Joni A. Cirelli, Fernando J. Von Zuben, Raquel Mantuaneli Scarel-Caminaga.

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
