## [Decision Letter · Decision Letter 0]

20 Jul 2020

PONE-D-20-19597

Using association rule mining to jointly detect clinical features and differentially expressed genes related to chronic inflammatory diseases

PLOS ONE

Dear Dr. Veroneze,

Thank you for submitting your manuscript to PLOS ONE. After careful consideration, we feel that it has merit but does not fully meet PLOS ONE’s publication criteria as it currently stands. Therefore, we invite you to submit a revised version of the manuscript that addresses the points raised during the review process.

Please respond point by point to all comments.

We look forward to receiving your revised manuscript.

Kind regards,

Paolo Magni

Academic Editor

PLOS ONE

Additional Editor Comments:

The Reviewers' comments need to be fully addressed.

2. We note that you are reporting an analysis of a microarray, next-generation sequencing, or deep sequencing data set. PLOS requires that authors comply with field-specific standards for preparation, recording, and deposition of data in repositories appropriate to their field. Please upload these data to a stable, public repository (such as ArrayExpress, Gene Expression Omnibus (GEO), DNA Data Bank of Japan (DDBJ), NCBI GenBank, NCBI Sequence Read Archive, or EMBL Nucleotide Sequence Database (ENA)). In your revised cover letter, please provide the relevant accession numbers that may be used to access these data. For a full list of recommended repositories, see http://journals.plos.org/plosone/s/data-availability#loc-omics or http://journals.plos.org/plosone/s/data-availability#loc-sequencing

3. We noted in your submission details that a portion of your manuscript may have been presented or published elsewhere:

'The gene expression dataset analyzed in our manuscript was also analyzed in the paper: Corbi et al., Scientific Reports (2020), which was properly cited in our manuscript. In this paper, the authors used traditional bioinformatic tools (Robust Multichip Average, RankProd, Ingenuity Pathway Analysis and Gene Set Enrichment Analysis) to identify differentially expressed genes. Patients' clinical features were not taken into account in the analysis this previous study.'

Please clarify whether this publication was peer-reviewed and formally published.

If this work was previously peer-reviewed and published, in the cover letter please provide the reason that this work does not constitute dual publication and should be included in the current manuscript.

Reviewers' comments:

Reviewer's Responses to Questions

**Comments to the Author**

1. Is the manuscript technically sound, and do the data support the conclusions?

Reviewer #1: Partly

Reviewer #2: Yes

2. Has the statistical analysis been performed appropriately and rigorously? 

Reviewer #1: Yes

Reviewer #2: Yes

3. Have the authors made all data underlying the findings in their manuscript fully available?

Reviewer #1: Yes

Reviewer #2: Yes

4. Is the manuscript presented in an intelligible fashion and written in standard English?

Reviewer #1: No

Reviewer #2: Yes

5. Review Comments to the Author

Reviewer #1: Dear Editor,

I carefully read the manuscript by Veroneze and collaborators, which regards and interesting and timely study.

My comments to improve the paper:

- English language is low-quality and needs to be carefully revised before resubmission.

- Table 1 - Classification of "Dyslipidemia" is substantially wrong and should be reformulated in accordance with the latest international guidelines

- References are obsolete and should be updated.

- The flowcharts are unclear.

Reviewer #2: Dear Authors, there are no doubts in the high actuality of the topic related to exploring an interaction patterns of type 2 diabetes mellitus, dyslipidemia and periodontitis based on its inflammatory background. The use of rule-based machine learning methods for identifying an interaction of clinical and molecular patients profiles adds an additional value. The manuscript is well written, clear and properly referenced.

There are no major issues that could affect the value of the research paper.

If to talk about minor issues, that probably could be taken into consideration for further scientific analysis are:

- relatively small size of the samples and the reasonability to increase the number of patients enrolled in order to get more convincing results for its extrapolation,

- probably, adding in the analysis patients group with type 2 DM and dyslipidemia with TG > 200 mg/ dL and decreased HDL-C (< 38 mg/dL and 46 mg/dL for men and women respectively) could be interesting in order to identify discrepancies with a group of patients with classic diabetic dyslipidemia. Moreover, not only quantitative, but qualitative changes in lipoproteins could be analyzed as well as adiponectin and leptin levels in order to better characterize clinical profiles.

6. PLOS authors have the option to publish the peer review history of their article (what does this mean?). If published, this will include your full peer review and any attached files.

Reviewer #1: No

Reviewer #2: No

---

## [Author Response · Author response to Decision Letter 0]

7 Sep 2020

All responses to editor's and reviewers’ comments and suggestions are available in the file 'ResponseToReviewersPONE-D-20-19597.pdf'

---

## [Decision Letter · Decision Letter 1]

23 Sep 2020

Using association rule mining to jointly detect clinical features and differentially expressed genes related to chronic inflammatory diseases

PONE-D-20-19597R1

Dear Dr. Veroneze,

We’re pleased to inform you that your manuscript has been judged scientifically suitable for publication and will be formally accepted for publication once it meets all outstanding technical requirements.

Kind regards,

Paolo Magni

Academic Editor

PLOS ONE

Additional Editor Comments (optional):

All comments have been properly addressed.

Reviewers' comments:

Reviewer's Responses to Questions

**Comments to the Author**

1. If the authors have adequately addressed your comments raised in a previous round of review and you feel that this manuscript is now acceptable for publication, you may indicate that here to bypass the “Comments to the Author” section, enter your conflict of interest statement in the “Confidential to Editor” section, and submit your "Accept" recommendation.

Reviewer #1: All comments have been addressed

Reviewer #2: (No Response)

2. Is the manuscript technically sound, and do the data support the conclusions?

Reviewer #1: Yes

Reviewer #2: Yes

3. Has the statistical analysis been performed appropriately and rigorously? 

Reviewer #1: Yes

Reviewer #2: Yes

4. Have the authors made all data underlying the findings in their manuscript fully available?

Reviewer #1: Yes

Reviewer #2: Yes

5. Is the manuscript presented in an intelligible fashion and written in standard English?

Reviewer #1: Yes

Reviewer #2: Yes

6. Review Comments to the Author

Reviewer #1: Dear Editor,

Authors revised their manuscript following my comments. I have no more suggestions on this interesting article.

Reviewer #2: The paper was harmonized based on the comments of reviewers. No additional comments within the second round review.

7. PLOS authors have the option to publish the peer review history of their article (what does this mean?). If published, this will include your full peer review and any attached files.

Reviewer #1: No

Reviewer #2: No

---

## [Editor Report · Acceptance letter]

24 Sep 2020

PONE-D-20-19597R1 

Using association rule mining to jointly detect clinical features and differentially expressed genes related to chronic inflammatory diseases 

Dear Dr. Veroneze:

I'm pleased to inform you that your manuscript has been deemed suitable for publication in PLOS ONE. Congratulations! Your manuscript is now with our production department. 

Kind regards, 

on behalf of

Prof. Paolo Magni 

Academic Editor

PLOS ONE